# Leading predictors and their associations with combination opioid pain therapy in older adults with cancer: Application of machine learning approaches

Christy Xavier[1], Rafia S. Rasu[1]*, Chanhyun Park[2], Sydney Manning[1], Usha Sambamoorthi[1]

1 Department of Pharmacotherapy, College of Pharmacy, University of North Texas Health Science Center, Fort Worth, Texas, United States of America, 2 Health Outcomes Division, College of Pharmacy, University of Texas at Austin, Austin, Texas, United States of America

* Rafia.Rasu@unthsc.edu

## Abstract

Combined use of opioids and other pharmacological therapies used for pain management, such as non-steroidal anti-inflammatory drugs (NSAIDs), benzodiazepines, gabapentinoids, and/or skeletal muscle relaxants (SMRs), in older adult cancer survivors can increase the risk of mortality. The objective of this study was to identify the leading predictors of opioid combination therapy with other therapies that may be inappropriate in older adults with cancer using interpretable machine learning approaches. A retrospective cohort design of older (> 66 years at diagnosis) cancer survivors (N = 2,682) diagnosed with primary and incident cancer in 2014. The Surveillance, Epidemiology, and End Results (SEER) cancer registry linked with Medicare claims database was used. Recursive feature elimination with random forest was used to extract the optimal number of features out of 119 for predictive modeling. The eXtreme Gradient Boosting (XGBoost), SHapley Additive exPlanations (SHAP), and global feature importance were used to identify the leading predictors and their associations with opioid combination therapy. Overall, 37.4% of older adults with cancer used opioid combination therapy. We included 34 features in the final predictive model. The predictive model had a good high area under the curve (AUC, 0. 758) and high recall (0.821) with the test dataset. We included 34 features in the final predictive model. Baseline opioids, NSAIDs, benzodiazepines, gabapentinoids, chemotherapy, surgery, and female sex generally positively predicted opioid combination therapy. We observed relationships of zip code percentage residents and Native American residents living below poverty with opioid combination therapy. Patient-level baseline medication use, biological factors, cancer treatment, and zip code-level poverty were leading predictors of opioid combination therapy. Our study findings contribute to the knowledge for targeted interventions to reduce the risk of opioid combination therapy.

**Data availability statement:** The SEER-Medicare 5% sample used in this analysis is a limited, de-identified dataset jointly owned by the National Cancer Institute (NCI) and the Centers for Medicare & Medicaid Services (CMS). Although all direct identifiers (e.g., names, social security numbers) are removed, the files include full-service dates and limited geographic variables (e.g., state, year/month of birth), so a remote risk of re-identification remains. To protect patient confidentiality, these data are not publicly available and may only be released under a strict Data Use Agreement (DUA) that prohibits redistribution and re-use beyond the approved project scope. Data Access Contact Requests for SEER-Medicare data (including execution of the DUA and submission of IRB documentation) should be directed to the SEER-Medicare program via: SEER-Medicare Program Information Management Services, Inc. 3901 Calverton Blvd., Suite 200 Calverton, MD 20705, USA E-mail: SEERMedicare@imsweb.com Web: https://healthcaredelivery.cancer.gov/seermedicare/obtain/.

**Funding:** The collection of cancer incidence data used in this study was supported by the California Department of Public Health pursuant to California Health and Safety Code Section 103885; Centers for Disease Control and Prevention's (CDC) National Program of Cancer Registries, under cooperative agreement 1NU58DP007156; the National Cancer Institute's Surveillance, Epidemiology and End Results Program under contract HHSN261201800032I awarded to the University of California, San Francisco, contract HHSN261201800015I awarded to the University of Southern California, and contract HHSN261201800009I awarded to the Public Health Institute. The ideas and opinions expressed herein are those of the author(s) and do not necessarily reflect the opinions of the State of California, Department of Public Health, the National Cancer Institute, and the Centers for Disease Control and Prevention or their Contractors and Subcontractors.

**Competing interests:** NO authors have competing interests.

## Introduction

Opioids remain a mainstay in pain management for cancer [1]. Opioid prescribing rates are much higher in patients with cancer than in the general population with pain [2]. Opioids are often used in combination with other analgesics and sedatives for pain management, but there are still concerns about the effectiveness and safety of concomitant use. Generally, older adults with chronic pain are prescribed non-steroidal anti-inflammatory drugs (NSAIDs) [3]. The combination of opioids and NSAIDs provides effective treatment for chronic pain, but this can lead to severe, even life-threatening adverse events such as gastrointestinal bleeding and renal dysfunction [4]. Furthermore, there is no robust evidence concerning the benefits of the combined use of opioids and NSAIDs given for cancer pain. In a meta-analysis, only nine of 14 trials demonstrated a slight benefit with the combination of opioids and NSAIDs for cancer pain [5]. In addition to NSAIDs, older adults with cancer have pre-existing chronic conditions that may warrant other types of medications with sedative properties, such as gabapentinoids, benzodiazepines, and skeletal muscle relaxants (SMRs). Older adults are often prescribed gabapentinoids, such as gabapentin and pregabalin, when they experience neuropathy due to chemotherapy [6]. Older adults have also been prescribed benzodiazepines for insomnia and anxiety, although clinical practice guidelines from several organizations advise against the use of benzodiazepines in older adults [7,8]. SMRs are also often used with opioids for the management of chronic pain, especially musculoskeletal pain [9,10]. These medications have been cautiously used due to their abuse potential and side effects.

The combined use of opioids with other pain medications can exacerbate adverse outcomes and increase the risk of mortality for opioids used with gabapentinoids [11], benzodiazepines [12,13], and SMRs [14]. For example, a population-based study found that the combined use of opioids and gabapentin was associated with a 49% increased risk of death from an opioid overdose [11]. In addition, combining opioids with gabapentinoids did not improve pain relief in patients with cancer compared with opioid monotherapy [15]. The combined use of opioids with NSAIDs, gabapentinoids, benzodiazepines, and/or SMRs may be considered potentially inappropriate medications (PIM) due to compromised patient safety.

More importantly, in real-world practice, the combined prescription of opioids and NSAIDs, gabapentinoids, benzodiazepines, and/or SMR is common in older adults with cancer. For example, in 2013, 69.0% of older adults with cancer were prescribed an opioid by an oncologist, but in response to the latest guidelines, this number has declined to 53.7% in 2017 [16]. Moreover, up to 80% of patients with cancer use adjunct non-opioid analgesics for pain control [17]. A previous study using the Surveillance, Epidemiology, and End Results (SEER) data linked with Medicare claims found that 25% of older adults with breast, lung, head and neck, or colorectal cancer received benzodiazepines during opioid therapy [18]. Another study reported that approximately 43% of patients with cancer were prescribed concurrent opioids with benzodiazepines or nonbenzodiazepine sedatives [19].

Despite the inappropriate use of opioid combination therapy with other therapies, little is known about the factors associated with those combination therapies in older adults with cancer. Therefore, the objective was to identify the leading predictors and their associations with opioid combination therapy in older adults with incident cancer using interpretable machine learning approaches.

## Materials and methods

### Study design

We used a retrospective longitudinal cohort design with baseline and follow-up periods. Baseline and follow-up periods were anchored to an index date, which was the cancer diagnosis date within the year 2014. The baseline was 12 months before cancer diagnosis; the follow-up period was 12 months after diagnosis (including the diagnosis month).

### Data sources

The data were derived from the SEER cancer registry linked with Medicare claims. SEER program is the largest and most authenticated source of information on patients with cancer in the United States. The SEER cancer registry contains data on cancer incidence, survival, burden, treatment, and demographics from 18 United States geographic areas since 1973. We used the 5% sample of cancer diagnosis file, which included individuals with cancer who resided in a SEER area and were in the Medicare 5% sample. We linked the SEER data with inpatient, outpatient, home health, prescription drug, durable medical equipment, and hospice claims provided by SEER. Zip code-level and census tract-level data were used to analyze population and socioeconomic characteristics.

This study was reviewed and approved by the Institutional Review Board of University of North Texas Health Science Center (Protocol # 1502351−5). This specific study was reviewed and approved by an institutional review board (North Texas Regional Institutional Review Board) before the study began. Reporting a retrospective study of medical records or archived samples, all data were fully anonymized before we accessed them and/or whether the IRB or ethics committee waived the requirement for informed consent. The need for informed consent was waived by the IRB committee.

### Study sample

Older (≥ 66 years old at diagnosis) cancer survivors diagnosed with primary and incident cancer in 2014 were selected for our study. We included the following cancers as provided in the SEER dataset: breast, bladder, cervical, colorectal, leukemia, lung, melanoma, non-Hodgkins lymphoma (NHL), ovarian, pancreatic, prostate, thyroid, and uterine cancers. Cancer types were identified by site codes provided by SEER. Although these cancers have differential physiologies and survival rates, opioids target the same receptor in all disease states, and therefore combining these cancers is not a limitation. We further restricted the cohort to those who had continuous Medicare Part A, B, and D programs. Medicare Part A provides inpatient/hospital coverage, Part B offers outpatient/medical coverage, and Part D is supplemental insurance that provides additional prescription drug coverage. Exclusion criteria included health maintenance organization (HMO) enrollment during the entire study period, and cancer diagnosis at autopsy. HMO refers to a type of health insurance plan that limits coverage to care from doctors who are contracted with that insurance company. These inclusion and exclusion criteria yielded a sample of 2,682 older adult cancer survivors (Table 1).

Overall, there were 2,682 who met all continuous enrollment and data-availability criteria—a 74.5% attrition overall. Attrition varied by cancer type, with the smallest losses in breast (61.7%) and prostate (62.9%) cancer patients and the greatest in lung cancer (89.0%). Much of this reduction reflects the strict 12-month fee-for-service (Parts A & B) and Part D enrollment requirements before and after diagnosis.

Table 1. Number of Individuals after inclusion and exclusion criteria SEER medicare database 5% sample 2013−2015[a].

| Reason for inclusion exclusion | ALL Cancers | Breast | Colorectal | Lung | Prostate | Other Cancers |
|---|---|---|---|---|---|---|
| | N | N | N | N | N | N |
| Diagnosed in 2014 | 10,502 | 1,748 | 1,192 | 1,924 | 1,503 | 4,135 |
| Primary cancer in 2014 | 7,660 | 1,362 | 836 | 1,371 | 1,239 | 2,852 |
| Excluded Diagnosed at Autopsy | 7,497 | 1,345 | 820 | 1,318 | 1,219 | 2,795 |
| Age >= 66 at diagnosis | 5,979 | 1,044 | 658 | 1,110 | 952 | 2,215 |
| 12-month enrollment Part A, B & D | 3,071 | 703 | 335 | 342 | 595 | 1,096 |
| 12-month enrollment Part D after Diagnosis | 2,682 | 670 | 288 | 212 | 557 | 955 |
| **Attrition Percentage** | **74.5%** | **61.7%** | **75.8%** | **89.0%** | **62.9%** | **76.9%** |

Based on older adults with breast, colorectal, lung, prostate, and other cancers. Other cancers included bladder, cervical, leukemia, melanoma, non-Hodgkins's lymphoma, ovarian, pancreatic, thyroid, and uterine cancers in 2014.

FFS: Fee-for-service; Dx: Diagnosis.

## Measures

### Target variable or primary outcome

The target (i.e., dependent or primary outcome) variable was opioid combination therapy during the 12 months after the incident of cancer diagnosis. The types of medications—NSAIDs, benzodiazepines, gabapentinoids, and/or SMRs— were identified using the generic name variable from Medicare Part D prescription drug event (PDE) files. Opioid combination therapy was defined as having an opioid prescription with any of NSAIDs, benzodiazepines, gabapentinoids, or SMR medications during the 12 months after a cancer diagnosis.

### Baseline features

We selected features based on the Anderson Behavioral Model of Health, published literature, and data available in the SEER-Medicare database. All features were measured during the 12 months before the cancer diagnosis. The features included a diverse set of socioeconomic, clinical, demographic, and healthcare factors grouped as predisposing, enabling, need, and healthcare utilization factors. Predisposing factors included individual-level factors such as age, sex, and race and ethnicity. Enabling factors included individual-level marital status at diagnosis, health insurance type, healthcare utilization, care fragmentation, dual Medicaid/Medicare eligibility, metro status, zip code-level education, and income.. Care fragmentation was calculated using a claims-based Fragmentation of Care Index (FCI) measuring the dispersion of care across multiple providers and specialties [20–22]. The measure varies from 0 to 100, with 100 representing the highest FCI. Need factors included coexisting individual-level type of cancer, stage from SEER registry, medical and mental health conditions identified from ICD-9 codes, and treatment was derived from procedure codes found in Medicare claims. Type of healthcare utilization consisted of inpatient, outpatient, and home health use.

### Descriptive analysis

Descriptive analysis included a description of individual-level characteristics of the analytical sample, prescription pain drug use before and after diagnosis of incident primary cancer, and group differences by opioid combination therapy after cancer diagnosis. Significant differences in pain drug use before and after diagnosis were tested with the McNemar test, a non-parametric method suitable for multiple observations on the same individual. Subgroup differences in opioid combination therapy after cancer diagnosis were tested using chi-square statistics.

## Machine learning analysis

We selected eXtreme Gradient Boosting (XGBoost) for classification to identify the leading predictors of opioid combination pain therapy use and to examine the associated features with combination opioid therapy. XGBoost utilizes gradient boosting, wherein it builds one tree at a time and considers the errors made by all previously built trees. This method promotes better predictive accuracy, multicollinearity, high-level interaction terms, and is better equipped to handle outliers than other algorithms. XGBoost is also known for faster processing speeds and regularization to prevent overfitting. The dataset was split into a training-validation set (80%) and a test set (20%) using stratified sampling to preserve the class distribution, with a random seed of 42 for reproducibility.

In our study, data analyses involved the following steps: data pre-processing, model development, model performance evaluation, feature importance derivation, and interpretation of results. Complex interactions were viewed for each variable by observing a positive or negative relationship between each variable with the target variable. All data pre-processing tasks were performed using SAS 9.4 (SAS Institute Inc. 2010, Cary, NC, USA). All ML analyses were conducted using Python 3.9.7 (Python Software Foundation, 2021) using the following libraries: pandas (data manipulation), scikit-learn (preprocessing, model evaluation), imblearn (oversampling), xgboost (classification), shap (interpretability), and matplotlib (visualization). The code was executed in an Anaconda environment with a consistent random seed, 42, for reproducibility.

**Data pre-processing.** In pre-processing, features were one-hot encoded, whereby categorical variables were reduced to a series of binary indicator variables for use in machine learning. Continuous features were standardized using StandardScaler from scikit-learn, while binary features were passed through unchanged. Our target variable was imbalanced (i.e., lower proportion of combination therapy users (37.4%) compared to non-combination therapy users (62.6%)) and to address class imbalance, Synthetic Minority Oversampling Technique for Nominal and Continuous features (SMOTENC) was applied to the training-validation set, generating synthetic positive samples to achieve a minority-to-majority ratio of approximately 0.697. SMOTENC preserved the categorical nature of binary features during oversampling. We also reduced the number of features using recursive feature elimination, known as RFE, with five-fold cross-validation using a random forest algorithm. We included the top 25 features identified through RFE in XGBoost analysis and included nine additional features that were not selected by RFE due to their associations with the target variable in previous literature.

**Model development.** We integrated pre-processing, oversampling, and classification and developed a XGBoost classifier. To further address class imbalance, the XGBoost classifier was configured with a scale_pos_weight parameter, set to the ratio of negative to positive samples (approximately 1.674). The model was optimized using RandomizedSearchCV with 100 iterations and 5-fold stratified cross-validation, maximizing the area under the receiver operating characteristic curve (ROC AUC). The hyperparameter search space included: number of trees (50–300), maximum tree depth (3–11), learning rate (0.01–0.3), subsample ratio (0.6–1.0), feature sampling percentage per tree (60%−100%), mimimum split loss (0–5), L1 regularization (0–2) and L2 regularization (0–2). The optimal hyperparameters were selected based on the highest cross-validated ROC AUC. To improve the reliability of predicted probabilities, the best model was calibrated using CalibratedClassifierCV with isotonic regression and 5-fold cross-validation. A custom decision threshold was determined by analyzing the precision-recall curve on the training-validation set; selecting the threshold that maximized recall while ensuring precision and recall both exceeded 0.6. If no such threshold existed, a default threshold of 0.5 was used.

**Model performance.** Model performance was evaluated on both the training-validation and test sets using multiple metrics such as accuracy (proportion of correct predictions), precision (proportion of positive predictions that were correct), recall (proportion of opioid combination therapy users correctly identified), area under the receiver operating curve (AUROC), measuring class separation, area under the precision-recall curve (AUPRC) that focused on positive class performance, average precision (weighted mean of precision at each recall level), log-loss (measure of probability prediction accuracy), and Brier score (mean squared error of predicted probabilities). Confusion matrices were generated

to visualize true positives, false positives, true negatives, and false negatives. AUROC curves were graphically represented to assess class discrimination.

**Model interpretability.** To improve the interpretability of the results from XGBoost and to summarize potential feature associations, we used SHapley Additive exPlanations (SHAP). [23] SHAP provides both feature importance and the contribution of each feature to the model prediction (global interpretability). With SHAP, feature contribution is derived by estimating the average marginal contribution of a specific feature to the model by using permutation and combinations of all features. Summary plots were used to display the top 10 features by mean absolute SHAP value. Partial Dependence Plots (PDPs) were created for these features, standardized to have consistent axes and a red horizontal line at zero to indicate baseline effects. SHAP interaction values were calculated using a TreeExplainer on a subset of 200 test samples to identify the top five feature interactions. Shapley values for our XGBoost model were obtained using the SHAP package in Python 3.9.7 (Python Software Foundation, the Netherlands).

## Results

### Sample characteristics

A description of the sample characteristics for adults included in our sample is presented in Table 2. Most of our sample were female (54.1%), Non-Hispanic White (74.5%), married (51.6%), and living in metropolitan areas (88.6%) The majority (29.3%) were between 70–74 years old, whereas a minority (20.3%) were between 75–79 years old. Breast (25.0%) and prostate (20.8%) cancers were the most common cancer types. The majority were in (stages 0–2, 75.2%), with most undergoing surgery after diagnosis (68.5%).

Table 3 shows pain prescription use before and after a cancer diagnosis. The proportion of older adults with pain prescriptions increased after cancer diagnosis, except for SMR. For example, the older adults with opioids, benzodiazepines, gabapentiooids, and NSAIDS increased from 31.5% to 68.9%, from 13.9% to 23.6%, 7.9% to 11.6%, and 22.3% to 22.9%, respectively.

Table 4 summarizes the characteristics of the sample by opioid combination pain therapy groups. Overall, 37.4% of older adults with cancer used opioid combination therapy. We observed significant differences in age at diagnosis, gender, race/ethnicity, marital status, Medicaid status, and non-malignant pain conditions by opioid combination therapy. For example, a higher percentage of Medicaid enrollees were on opioid combination therapy compared to those without Medicaid (51.3% vs. 36.6%). In addition, cancer type, cancer stage, and cancer treatment were also associated with the use of opioid combination therapy. A higher percentage of older adults with lung (49.5%) or breast (46.9%) cancers, stage 3 cancer (47.0%), and chemotherapy (52.8%) had opioid combination therapy compared to their counterparts.

### Model performance

On the training + validation dataset, the model achieved an accuracy of 73.5%, with a precision of 60.1% and a recall of 86.8%, indicating strong identification of positive cases (Table 5). The ROC AUC was 85.7%, suggesting robust class separation, while the area under the precision-recall curve (AUPR) and average precision were 79.0% and 78.8%, respectively, demonstrating effective positive class performance. On the test set, accuracy was 66.3%, with a precision of 53.3% and a recall of 79.6%, showing good generalization but a slight drop in precision. The test ROC AUC was 76.0%, with AUPR and average precision at 65.4% and 65.4%, respectively. Probability calibration achieved a log loss of 0.475 (training + validation dataset) and 0.564 (test dataset), and Brier scores of 0.155 and 0.191, indicating reliable probability estimates.

### Leading predictors and feature contributions to predictions

We observed that benzodiazepine, NSAID, and opioid use before diagnosis, age at diagnosis, female sex, gabapentinoid use before diagnosis, surgery, chemotherapy, female sex, percent of residents living in poverty, percent of

**Table 2. Description of selected characteristics among older adults with incident cancer in 2014: SEER medicare database 5% sample 2013–2015.**

|  | N | % |
|---|---|---|
| **Age at Diagnosis** | | |
| 66-69 | 729 | 27.2 |
| 70-74 | 786 | 29.3 |
| 75-79 | 544 | 20.3 |
| >80 | 623 | 23.2 |
| **Sex** | | |
| Female | 1,451 | 54.1 |
| Male | 1,231 | 45.9 |
| **Race and Ethnicity** | | |
| Hispanic | 1,998 | 74.5 |
| Non-Hispanic Black | 201 | 7.5 |
| Non-Hispanic White | 218 | 8.1 |
| Other | 194 | 7.2 |
| **Marital Status** | | |
| Married | 1,384 | 51.6 |
| Widowed/Separated/Divorced | 732 | 27.3 |
| Never Married | 237 | 8.8 |
| **Medicaid** | | |
| Yes | 154 | 5.7 |
| No | 2,528 | 94.3 |
| **Private Insurance** | | |
| Yes | 754 | 28.1 |
| No | 1,928 | 71.9 |
| **Other Insurance** | | |
| Yes | 43 | 1.6 |
| No | 2,639 | 98.4 |
| **Metro** | | |
| Yes | 2,376 | 88.6 |
| No | 306 | 11.4 |
| **Region** | | |
| Northeast | 463 | 17.3 |
| South | 558 | 20.8 |
| North Central | 289 | 10.8 |
| West | 1,372 | 51.2 |
| **Non-malignant chronic pain conditions** | | |
| Yes | 1,053 | 39.3 |
| No | 1,629 | 60.7 |
| **Cancer Type** | | |
| Breast | 670 | 25.0 |
| Colorectal | 288 | 10.7 |
| Lung | 212 | 7.9 |
| Prostate | 557 | 20.8 |
| All Others | 955 | 35.6 |
| Cancer Stage | | |
| Stage 0 | 417 | 15.5 |

*(Continued)*

**Table 2.** (Continued)

|  | N | % |
|---|---|---|
| Stage 1 | 803 | 29.9 |
| Stage 2 | 800 | 29.8 |
| Stage 3 | 249 | 9.3 |
| Stage 4 | 189 | 7.0 |
| **Cancer Treatment after diagnosis** | | |
| Radiation | 764 | 28.5 |
| Surgery | 1,838 | 68.5 |
| Chemotherapy | 354 | 13.2 |

Based on 2,682 older adults aged 66 and older with primary incident breast, colorectal, lung, prostate, bladder, cervical, leukemia, melanoma, non-Hodgkins lymphoma, ovarian, pancreatic, thyroid, and uterine cancers diagnosed in 2014, continuously enrolled in fee-for-service Medicare parts A, B, and D throughout the study period, and not diagnosed with cancer at autopsy.

**Table 3.** Pain prescription use before and after diagnosis of cancer among older adults with incident cancer in 2014: SEER medicare database 5% sample 2013–2015.

| Medication | 12 m before diagnosis | | 12 m after diagnosis | | Change in % Points | p-value |
|---|---|---|---|---|---|---|
|  | N | % | N | % |  |  |
| Opioid | 846 | 31.5 | 1,847 | 68.9 | +37.4 | <.001 |
| Benzodiazepine | 374 | 13.9 | 633 | 23.6 | +9.7 | <.001 |
| Gabapentinoid | 213 | 7.9 | 312 | 11.6 | +3.7 | <.001 |
| NSAID | 597 | 22.3 | 615 | 22.9 | +0.6 | 0.4780 |
| SMR | 143 | 5.3 | 134 | 5.0 | −0.3 | 0.5432 |

Based on 2,682 older adults aged 66 and older with primary incident breast, colorectal, lung, prostate, bladder, cervical, leukemia, melanoma, non-Hodgkins lymphoma, ovarian, pancreatic, thyroid, and uterine cancers diagnosed in 2014, continuously enrolled in fee-for-service Medicare parts A, B, and D throughout the study period, and not diagnosed with cancer at autopsy. Abbreviation: NSAIDs - Non-steroidal anti-inflammatory drugs; SMR: skeletal muscle relaxants.

P-values were derived from the McNemar non-parametric test.

Native Americans living in poverty, and pain conditions (Fig 1a). Fig 1b shows how features contribute to model predictions, showing the magnitude and direction of each feature's impact on the log-odds of predicting opioid combination therapy after diagnosis. This plot highlights the model's decision-making process, complementing performance metrics. Benzodiazepine, NSAID, opioid use before diagnosis, female sex, surgery, gabapentinoid use, chemotherapy, and pain conditions had a positive relationship with opioid combination therapy. There were complex associations with age, percent living below poverty, and percent Native American residents living below poverty with opioid combination therapy use.

The age partial-dependence curve shows a non-linear relationship; predicted risk is highest for younger groups, declining steadily with each additional year, with attenuated risk in the very old cancer survivors (Fig 2). In zip codes where 0–10% and more than 40% living below poverty, there appeared to be the highest predictive value of opioid combination use. Similarly, in zip codes where 20%−60% of Native Americans living in poverty, the model predicted above-average risk of opioid combination therapy after diagnosis. As a continuous variable, age has a seemingly negative relationship with opioid combination therapy. Younger age was predictive of opioid combination therapy, whereas older age was generally associated with no opioid combination therapy use.

**Table 4. Characteristics by opioid combination therapy after cancer diagnosis among older adults with incident cancer in 2014: SEER medicare database 5% sample 2013–2015.**

| | Opioid Combination Therapy | | No Opioid Combination Therapy | | Chi-sq value | P-value[d] |
|---|---|---|---|---|---|---|
| | *N* | *%* | *N* | *%* | | |
| **All** | 1,003 | 37.4 | 1,679 | 62.6 | | |
| **Age at Diagnosis** | | | | | | |
| 66-69 | 324 | 44.4 | 405 | 55.6 | 40.42 | < 0.001 |
| 70-74 | 301 | 38.3 | 485 | 61.7 | | |
| 75-79 | 205 | 37.7 | 339 | 62.3 | | |
| ≥80 | 173 | 27.8 | 450 | 72.2 | | |
| **Sex** | | | | | | |
| Female | 661 | 45.6 | 790 | 54.4 | 89.85 | < 0.001 |
| Male | 342 | 27.8 | 889 | 72.2 | | |
| **Race and Ethnicity** | | | | | | |
| Hispanic | 93 | 42.3 | 125 | 57.3 | 23.77 | < 0.001 |
| Non-Hispanic Black | 89 | 44.3 | 112 | 55.7 | | |
| Non-Hispanic White | 753 | 37.7 | 1,245 | 62.3 | | |
| Other | 54 | 27.8 | 140 | 55.7 | | |
| **Marital Status** | | | | | | |
| Married | 506 | 36.6 | 878 | 63.4 | 36.07 | < 0.001 |
| Widowed/Separated/Divorced | 313 | 42.8 | 419 | 57.2 | | |
| Never Married | 103 | 43.5 | 134 | 56.5 | | |
| **Medicaid** | | | | | | |
| Yes | 79 | 51.3 | 75 | 48.7 | 13.49 | 0.0002 |
| No | 924 | 36.6 | 1604 | 63.4 | | |
| **Private Insurance** | | | | | | |
| Yes | 296 | 39.3 | 458 | 60.7 | 1.55 | 0.2132 |
| No | 707 | 36.7 | 1221 | 63.3 | | |
| **Other Insurance** | | | | | | |
| Yes | 11 | 25.6 | 32 | 74.4 | 2.61 | 0.1064 |
| No | 992 | 37.6 | 1647 | 62.4 | | |
| **Metro** | | | | | | |
| Yes | 888 | 37.4 | 1488 | 62.6 | 0.01 | 0.9436 |
| No | 115 | 37.6 | 191 | 62.4 | | |
| **Region** | | | | | | |
| Northeast | 160 | 34.6 | 303 | 65.4 | 19.66 | 0.0002 |
| South | 251 | 45.0 | 307 | 55.0 | | |
| North Central | 92 | 31.8 | 197 | 68.2 | | |
| West | 500 | 36.4 | 872 | 63.6 | | |
| **Non-malignant pain conditions** | | | | | | |
| Yes | 473 | 44.1 | 580 | 55.1 | 25.81 | <0.001 |
| No | 530 | 32.5 | 1,099 | 67.5 | | |
| **Cancer Type** | | | | | | |
| Breast | 314 | 46.9 | 356 | 53.1 | 70.67 | < 0.001 |
| Colorectal | 98 | 34.0 | 190 | 66.0 | | |
| Lung | 105 | 49.5 | 107 | 50.5 | | |
| Prostate | 147 | 26.4 | 410 | 73.6 | | |

*(Continued)*

**Table 4.** (Continued)

| | Opioid Combination Therapy | | No Opioid Combination Therapy | | Chi-sq value | P-value[d] |
|---|---|---|---|---|---|---|
| | N | % | N | % | | |
| All Others | 339 | 35.5 | 616 | 64.5 | | |
| **Cancer Stage** | | | | | | |
| Stage 0 | 118 | 28.3 | 299 | 71.7 | 42.31 | < 0.001 |
| Stage 1 | 349 | 43.5 | 454 | 56.5 | | |
| Stage 2 | 269 | 33.6 | 531 | 66.4 | | |
| Stage 3 | 117 | 47.0 | 132 | 53.0 | | |
| Stage 4 | 76 | 40.2 | 113 | 59.8 | | |
| **Cancer Treatment** | | | | | | |
| Radiation | 320 | 41.9 | 444 | 58.1 | 9.30 | 0.0023 |
| Surgery | 751 | 40.9 | 1087 | 59.1 | 29.90 | < 0.001 |
| Chemotherapy | 187 | 52.8 | 167 | 47.2 | 41.46 | < 0.001 |

Based on 2,682 older adults aged 66 and older with primary incident breast, colorectal, lung, prostate, bladder, cervical, leukemia, melanoma, non-Hodgkins lymphoma, ovarian, pancreatic, thyroid, and uterine cancers diagnosed in 2014, continuously enrolled in fee-for-service Medicare parts A, B, and D throughout the study period, and not diagnosed with cancer at autopsy.

Statistical significance was tested using Chi-Square analysis.

**Table 5. Predictive model performance of combination opioid therapy: Older adults with incident cancer in 2014: SEER medicare database 5% sample 2013–2015.**

| Metrics | Training + Validation | Test |
|---|---|---|
| AUROC | 0.864 | 0.758 |
| Recall | 0.874 | 0.821 |
| Accuracy | 0.735 | 0.659 |
| Precision | 0.600 | 0.529 |
| AUPRC | 0.807 | 0.660 |
| Average Precision | 0.806 | 0.661 |
| Log Loss | 0.472 | 0.561 |
| Birer Score | 0.153 | 0.190 |

Based on 2,682 older adults aged 66 and older with primary incident breast, colorectal, lung, prostate, bladder, cervical, leukemia, melanoma, non-Hodgkins lymphoma, ovarian, pancreatic, thyroid, and uterine cancers diagnosed in 2014, continuously enrolled in fee-for-service Medicare parts A, B, and D throughout the study period, and not diagnosed with cancer at autopsy.

## SHAP feature-feature interactions

The top SHAP interactions included surgery status and benzodiazepine use (0.0278 log-odds), percent below poverty American Indian and other groups (0.0269), and fragmented care index with percent Native Hawaiians living below poverty (0.0253), benzodiazepine use before diagnosis and percent American Indian (0.0215) living below poverty and percent other race groups living below poverty and fragmented care index (0.0211), highlighting synergistic effects in predicting opioid combination therapy.

## Discussion

We found that seven in 10 adults used opioids with and without combination therapies after cancer diagnosis suggesting that opioids continue to play a central role in cancer pain treatment [1]. Despite the risk of negative health outcomes

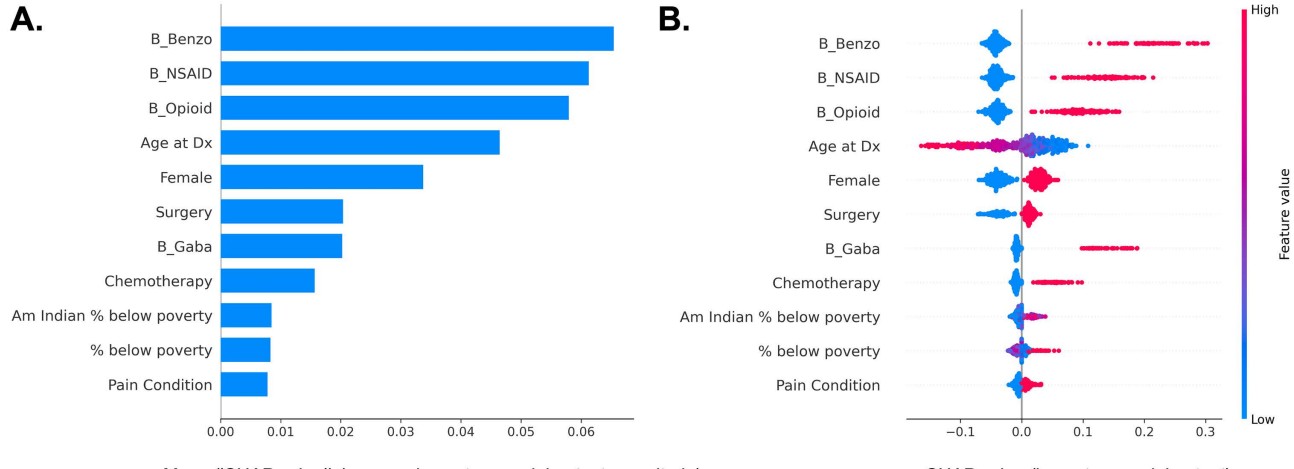

**Fig 1. SHAP summary plot of top 10 leading predictors of opioid combination therapy after diagnosis. A.** In the figure x-axis represents absolute SHAP value for each feature, which quantifies the average magnitude of each feature's impact on the model's predictions across all samples in the test set. **B.** The x-axis represents the marginal contribution of a feature to the change in the predicted probability of opioid combination therapy; Negative values indicate a negative association with opioid combination therapy, while positive values indicate an increased likelihood of opioid combination therapy. Red color indicates high feature values, whereas blue color indicates low. If red dots (e.g., feature value = 1) cluster to the right, then it suggests that feature value = 1 increases the log-odds of combination opioid therapy after diagnosis. If blue dots (e.g., feature value = 0) cluster to the right then feature value = 0 increases the log-odds of combination opioid therapy. A wide spread of dots indicates the feature has varied impacts across samples. The x-axis represents the marginal contribution of a feature to the change in the predicted probability of opioid combination therapy; red color represents increase, and blue color represents decrease in the incidence of opioid combination therapy. Negative values indicate a negative association with opioid combination therapy, while positive values indicate an increased likelihood of opioid combination therapy. Benzodiazepine, NSAID, opioid use before diagnosis, female sex, surgery, gabapentinoid use, chemotherapy, and pain conditions had a positive relationship with opioid combination therapy. There were complex associations with age, percent living below poverty, and percent Native American residents living below poverty with opioid combination therapy use. NSAID: non-steroidal anti-inflammatory drug, Dx: diagnosis, Am Indian = American Indian; B_: Before cancer diagnosis.

among older cancer survivors, combined use of opioids and other therapies is still common among older adults with cancer. Cancer pain is a complex process relating to cancer cells, the nervous system, and the immune system [24].

In this nationwide sample of cancer survivors, approximately 37% of older adults with cancer were prescribed opioids in combination with NSAID, benzodiazepine, gabapentinoid, and/or SMRs. This finding is consistent with those of previous research. In the U.S. population level study using the SEER-Medicare data, 25% of older adults with breast, lung, head and neck, or colorectal cancer were prescribed concurrent opioids with benzodiazepines [18]. Previous studies at an outpatient palliative care clinic reported that approximately 43% of patients with various cancers were co-prescribed opioids and benzodiazepines/nonbenzodiazepine sedatives [19], and 49% of patients with cancer receiving opioids were also used gabapentinoids [25]. As such, identification of a sub-group with a higher risk of the concurrent use of opioids with other therapies is critical, so that practitioners pay increasing attention to the prevention of this misuse of opioids.

More importantly, we found that the leading predictors of combination opioid use after diagnosis were baseline pain medication prescriptions, age, female, cancer treatment, such as surgery and chemotherapy, percent of residents living in poverty, and percent of Native Americans living in poverty. The use of benzodiazepines, NSAIDs, and gabapentinoids before cancer diagnosis strongly predicted opioid combination therapy use after a cancer diagnosis. It is possible that individuals who had these prescriptions before were most likely to continue them upon diagnosis as part of their disease management. A national Norweigan cohort study showed that the use of analgesic adjuncts persisted in older cancer survivors ten years post-diagnosis, and this resulted in higher opioid and analgesic doses than the general population [26].

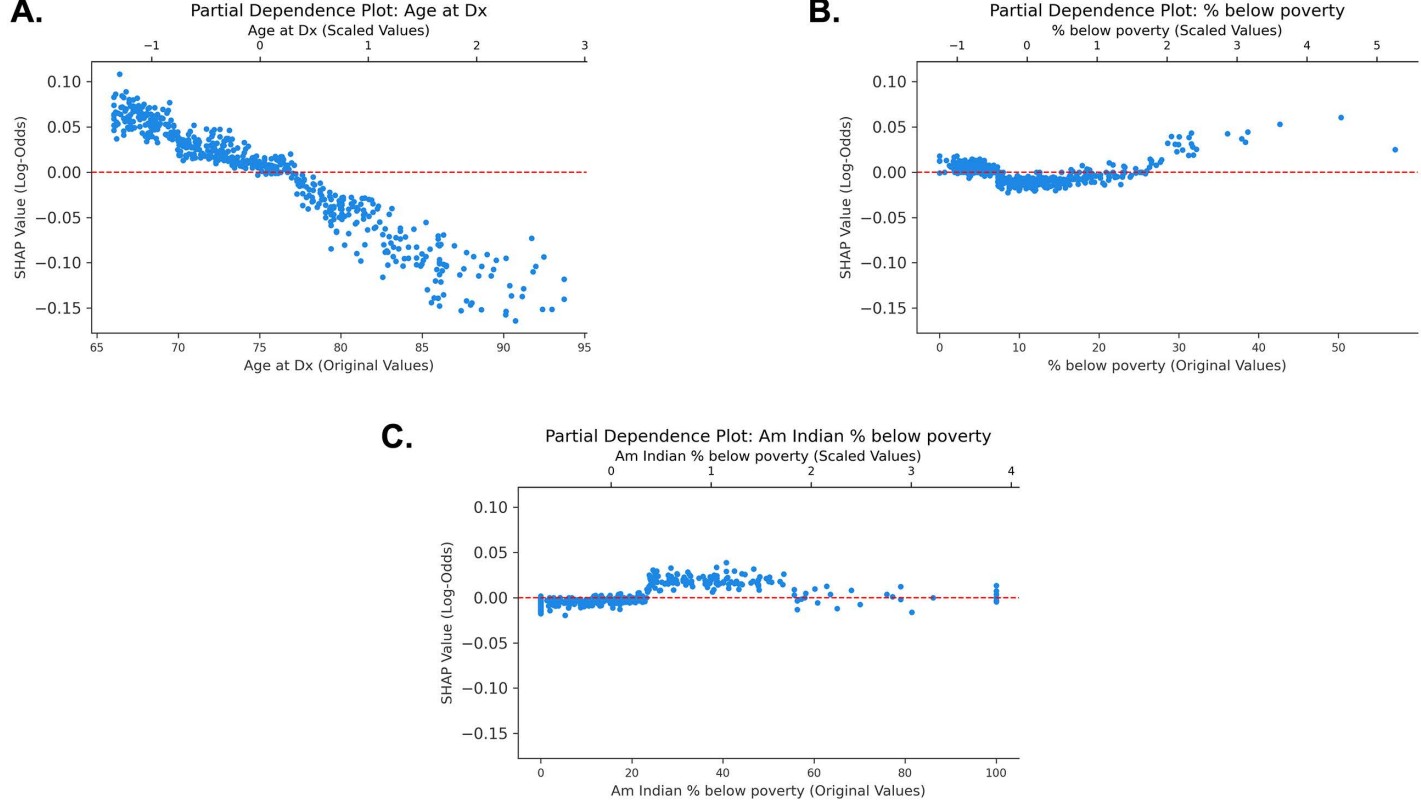

**Fig 2. Partial dependence plots of age, areas with percent living below poverty, and areas with percent of Native Americans living below poverty. A.** The partial dependence plot of age at diagnosis. **B.** The partial dependence plot of percent residents living below poverty. **C.** The partial dependence plot of percent Native Americans living in poverty. Dx=diagnosis, Am Indian=American Indian.

In our study, cancer treatment, such as chemotherapy and surgery, were two of the top ten predictors of opioid combination therapy use after a cancer diagnosis, suggesting appropriate management of pain. Prior studies support this finding. For example, one study examining prolonged opioid use after mastectomy showed that adjunct chemotherapy as well as perioperative opioid use both predicted long-term opioid use in breast cancer [27]. Another study analyzed persistent opioid use 3 and 6 months after curative treatment in stage III breast patients with cancer and showed a 44% higher chance of chronic opioid use with surgery and chemotherapy [28].

We found neighborhood characteristics such as percent Native Americans living below poverty and percent residents living below poverty to be leading predictors of opioid combination therapy. This finding is consistent with a study on concomitant opioid and benzodiazepine use in patients with breast cancer that reported a high prevalence of combination therapy in low-income and rural communities [29]. Interestingly, opioid use is highly prevalent among American Indians and Alaska Natives [30–32]. The opioid crisis in this population probably also leads to a high prevalence of using opioids with other medications.

The findings from this study have important implications for clinicians, policymakers, and other stakeholders for pain management. For example, cancer survivors with diagnosed pain conditions were more likely to receive combination opioid prescriptions. There was an interactive effect of surgery and benzodiazepine use on prediction combination opioid therapy, possibly due to complex pain management needs post-surgery. However, co-prescription of opioids and benzodiazepines is a known risk factor for adverse events (e.g., respiratory depression, overdose). All these findings highlight high-risk subgroups of cancer survivors for intervention.

Although current guidelines recommend against opioid combination therapy unless warranted [8], our study using real-world data has revealed a substantial prevalence of opioid combination therapy in older cancer survivors. Understanding the predictors of combination therapy used in this population can increase awareness among clinicians of at-risk patients. Our findings also suggest the need for collaborative care and pharmacist-led interventions because of a high prevalence of pre-existing chronic conditions among patients with cancer. Numerous studies found that pharmacist-led interventions have proven to be successful in reducing inappropriate prescription combinations in older adults [33,34]. Applying STOPP (Screening Tool of Older Persons' Prescriptions) and START (Screening Tool to Alert to Right Treatment) tools to this population through collaborative care can help reduce opioid combination therapy and its associated risks [35]. Medication therapy management (MTM) can be effective in identifying common drug-related problems among older patients with cancer [36]. These tools and programs can be effectively used for the prevention and management of opioid misuse.

There are several study limitations. First, our analysis may not be generalizable to all older adults because it excluded Medicare beneficiaries enrolled in health maintenance organization (HMO) and those who did not have Medicare Part D coverage. We did not include over-the-counter (OTC) pain medications or other types of pain management use. Furthermore, Medicare Part D files contain only filled prescriptions, and we do not reflect the actual use of medications. Insurance variables were derived from the variable primary payer at diagnosis, a common limitation reported for cancer registry primary payer information impacting insurance-related policy [37]. We did not assess the severity of pain; however, we adjusted for the presence of chronic pain conditions before the diagnosis of cancer. Nevertheless, this study has several strengths. First, this is the first nationwide cohort study to analyze the prevalence and predictors of combination pain therapy in older adults with cancer. This study also uses real-world data from a cancer registry that is nationally representative. We also used machine learning methods including interpretable machine learning algorithms to identify predictors and explain associations.

## Conclusion

In conclusion, approximately one in three older cancer survivors in our study were on opioid combination therapy. The most influencing predictors of combination therapy use were previous pain medication prescriptions, suggesting that surveillance and close monitoring of older adults with past pain medication use is warranted.

## Author contributions

**Conceptualization:** Rafia S. Rasu, Christy Xavier, Usha Sambamoorthi.

**Data curation:** Rafia S. Rasu, Usha Sambamoorthi.

**Formal analysis:** Christy Xavier, Sydney Manning.

**Investigation:** Rafia S. Rasu.

**Methodology:** Sydney Manning, Usha Sambamoorthi.

**Supervision:** Rafia S. Rasu.

**Writing – original draft:** Rafia S. Rasu, Christy Xavier, Chanhyun Park, Usha Sambamoorthi.

**Writing – review & editing:** Chanhyun Park.

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
