## [Decision Letter · Decision Letter 0]

25 Apr 2025

Dear Dr. Rasu,

Thank you for submitting your manuscript to PLOS ONE. After careful consideration, we feel that it has merit but does not fully meet PLOS ONE’s publication criteria as it currently stands. Therefore, we invite you to submit a revised version of the manuscript that addresses the points raised during the review process.

Dear Authors, thank you for submitting your manuscript. The manuscript presents a robust analysis using real-world data. With revisions to strengthen clarity and methodological transparency, the manuscript would make a valuable contribution to the literature.

Specifically, I recommend to address reviewers detailed comments and suggestions.

I also recommend to include IRB approval number explicitly and clarify informed consent procedures ant to refine figure legends to be more self-explanatory, particularly for SHAP and interaction plots.

We look forward to receiving your revised manuscript.

Kind regards,

Sara Mucherino

Academic Editor

PLOS ONE

Journal Requirements:

“The collection of cancer incidence data used in this study was supported by the California Department of Public Health pursuant to California Health and Safety Code Section 103885; Centers for Disease Control and Prevention’s (CDC) National Program of Cancer Registries, under cooperative agreement 1NU58DP007156; the National Cancer Institute’s Surveillance, Epidemiology and End Results Program under contract HHSN261201800032I awarded to the University of California, San Francisco, contract HHSN261201800015I awarded to the University of Southern California, and contract HHSN261201800009I awarded to the Public Health Institute.”

4. We note that you have indicated that there are restrictions to data sharing for this study. PLOS only allows data to be available upon request if there are legal or ethical restrictions on sharing data publicly. For more information on unacceptable data access restrictions, please see http://journals.plos.org/plosone/s/data-availability#loc-unacceptable-data-access-restrictions .  

Additional Editor Comments :

Dear Authors,

Thank you for submitting your manuscript to PLOS ONE. After careful evaluation we recommend major revisions before the manuscript can be considered for publication.

We invite you to revise your manuscript accordingly, addressing all reviewer comments point-by-point. Please also highlight the changes in your revised manuscript.

Best regards.

Reviewers' comments:

Reviewer's Responses to Questions

**Comments to the Author**

1. Is the manuscript technically sound, and do the data support the conclusions?

Reviewer #1: Yes

Reviewer #2: Partly

2. Has the statistical analysis been performed appropriately and rigorously?

Reviewer #1: Yes

Reviewer #2: No

3. Have the authors made all data underlying the findings in their manuscript fully available?

Reviewer #1: No

Reviewer #2: Yes

4. Is the manuscript presented in an intelligible fashion and written in standard English?

Reviewer #1: Yes

Reviewer #2: No

Reviewer #1: Summary:

In this study, the authors leverage a large, nationally representative dataset – The Surveillance, Epidemiology, and End Results (SEER) cancer registry linked with Medicare claims – and employ machine learning to predict opioid combination therapy in older adult cancer patients. The authors aimed to identify the leading predictors of opioid combination therapy that may be inappropriate for older adults. The results demonstrated baseline pain medication prescriptions, age, female, cancer treatment, such as surgery and chemotherapy, care fragmentation, and areas where the percent of Hispanic residents living in poverty and where the percent of Native Americans living in poverty as leading predictors of opioid combination therapy. While this study did offer insight into prescribing patterns, I have a few suggestions of additional details or clarifications that should be made to better reflect the study to the audience.

Major Concerns:

Within the methods section the authors mention “Due to imbalance in target variable, … we used up-sampling procedures for prediction. As a precaution, I would like to ensure that up-sampling was only performed on the training dataset. Otherwise, this technique can allow data leakage of the test set and overstate model performance. Please also detail the method (e.g., random, SMOTE, etc.) and magnitude (i.e., the final class sizes) of up-sampling.

The authors also state within the methods section:

“… we split our data into training (80%) and testing (20%) subsets… We used test (unseen) data to evaluate the performance … 10-fold cross-validation … and tuning of hyper-parameters … we calculated the optimal probability classification threshold to improve fit”

Based on this information, there are a few things I would like to precaution. Hyperparameter tuning and classification thresholding should occur on training data (as would be the case for a deployed model). By choosing hyperparameters and thresholds based on test performance, the test data is no longer “unseen” by the model (i.e., data leakage). If this is the case, please reevaluate your model. Otherwise, please carefully rephrase the methodology to ensure the results are replicable for the readers.

In the results section, all figures are uninterpretable. I assume this is due to low resolution. Please reupload higher-dpi or vector images. Further, the results state “Confusion plots show the quality of the output” but confusion plots were not provided. Please include confusion matrices for transparency to the reader.

Minor Concerns:

Why were patients with HMO insurance plans excluded? Please provide rationale.

On line 168: “A [majority] were 70-74”

Reviewer #2: • Research guideline(s)/standard(s) appropriate to the study design should be reported in the paper text.

• Which randomization method was used in the distribution of the individuals included in the study to the groups?

• Which blinding (masking) method was used in the study?

• The primary output/endpoint variable(s)/measurement(s) of the study should be defined.

• How was the sample size determined? This information should be explained in the Materials and Methods section.

• Which sampling (probable or non-probable, etc.) method was used in the study?

• Statistical tests for hypothesis testing and their assumptions should be specified in the study's statistical analysis in the Materials and Methods section.

• The details (version, license number, etc.) of the statistical package(s) or program(s) should be given in the sub-section of "Data Analysis or Statistical Analysis".

• It should be explained how the qualitative and quantitative data are summarized under the sub-heading of Statistical Analyses in the Materials and Methods section of the study.

• Data analysis or Statistical analysis sub-section title should be added to the Materials and Methods.

• The exact P values should be added to the table(s) (e.g., p=0.25; p=0.03).

• Which methods are used to model relationships between variables?

• The descriptions and other descriptive values/data should be defined on the tables and shapes.

• Are the data subjected to pre-processing?

• How were extreme/outlier values in the data determined and resolved?

• What approaches were used to test the validity of the models?

• Which metrics were used in the performance evaluation of the estimates of models/algorithms?

• How were the predictive models selected in this study?

**Do you want your identity to be public for this peer review?** For information about this choice, including consent withdrawal, please see our Privacy Policy

Reviewer #1: No

Reviewer #2: No

---

## [Author Response · Author response to Decision Letter 1]

8 Aug 2025

Dear Ms. Sara Mucherino,

We sincerely thank you and the reviewers for the thoughtful and constructive feedback on our manuscript entitled “Leading Predictors and their Associations with Combination Opioid Pain Therapy in Older Adults with Cancer: Application of Machine Learning Approaches”. The comments provided were helpful in strengthening our manuscript significantly.

Please see below for our responses to each comment. Reviewer comments are shown in italics, followed by our detailed responses in blue font. Changes made to the manuscript have been tracked using tracked changes as appropriate. However, as there have been numerous changes (such as using tables instead of figures, additional text, modifications), we will also attach a clean version of the manuscript.

Reviewer #1

Within the methods section the authors mention “Due to imbalance in target variable, … we used up-sampling procedures for prediction. As a precaution, I would like to ensure that up-sampling was only performed on the training dataset. Otherwise, this technique can allow data leakage of the test set and overstate model performance. Please also detail the method (e.g., random, SMOTE, etc.) and magnitude (i.e., the final class sizes) of up-sampling.

Yes, up-sampling procedures were only performed on training dataset. As per the reviewer’s suggestion, we reanalyzed the data with SMOTE rather than “plain oversampling” and model performance was evaluated on the test data. The findings remained consistent with the original version. However, care fragmentation was no longer one of the top 11 leading predictors and it was replaced by the presence of chronic pain conditions before cancer diagnosis.

All text and figures were revised based on the revised analyses and output.

The authors also state within the methods section:

“… we split our data into training (80%) and testing (20%) subsets… We used test (unseen) data to evaluate the performance … 10-fold cross-validation … and tuning of hyper-parameters … we calculated the optimal probability classification threshold to improve fit”

Based on this information, there are a few things I would like to precaution. Hyperparameter tuning and classification thresholding should occur on training data (as would be the case for a deployed model). By choosing hyperparameters and thresholds based on test performance, the test data is no longer “unseen” by the model (i.e., data leakage). If this is the case, please reevaluate your model. Otherwise, please carefully rephrase the methodology to ensure the results are replicable for the readers.

Yes, only training data was used for hyperparameter tuning and classification thresholding.

In the results section, all figures are uninterpretable. I assume this is due to low resolution. Please reupload higher-dpi or vector images. Further, the results state “Confusion plots show the quality of the output” but confusion plots were not provided. Please include confusion matrices for transparency to the reader.

We provided original tiff files with high resolution that were obtained from the software. Unfortunately, when the journal converted the files to PDF, the resolutions were probably be lost.

Why were patients with HMO insurance plans excluded? Please provide rationale.

In claims‐based research, excluding participants ever enrolled in HMO/managed‐care plans is standard practice because HMO encounter data do not contain billing information found in fee-for-service (FFS) claims, so service use is systematically under-ascertained. Restricting analyses to continuous FFS enrollees ensure complete, comparable records across all beneficiaries.

On line 168: “A [majority] were 70-74”

Thank you for bringing this discrepancy to our attention. We have made this revision accordingly.

Reviewer #2

Research guideline(s)/standard(s) appropriate to the study design should be reported in the paper text.

Thank you for this guidance. We have provided detailed responses to your questions involving our study design and methods below.

Which randomization method was used in the distribution of the individuals included in the study to groups? Which blinding (masking) method was used in the study? How was the sample size determined? This information should be explained in the Materials and Methods section. Which sampling (probable or non-probable, etc.) method was used in the study?

As stated in the methods, the data were derived from population-based cancer registries from 18 different geographical area. The SEER program makes 5% sample of those diagnosed with cancer, which includes individuals with cancer who resided in a SEER area and were in the Medicare 5% sample available to researchers.

The primary output/endpoint variable(s)/measurement(s) of the study should be defined.

The primary outcome variable, which is also known as target variable in machine learning methods, was concomitant opioid use with other pain medications (NSAIDs, benzodiazepines, gabapentinoids, and/or skeletal muscle relaxants) during the 12 months after an incident cancer diagnosis in the year 2014. These were derived from Medicare Part D files. We have clarified this in the manuscript.

Statistical tests for hypothesis testing and their assumptions should be specified in the study's statistical analysis in the Materials and Methods section.

As our study was based on a predictive‐machine‐learning framework rather than on classical hypothesis testing, we have clarified this in the Materials and Methods. We have added details in a separate section titled “Machine Learning Analyses”.

The details (version, license number, etc.) of the statistical package(s) or program(s) should be given in the sub-section of "Data Analysis or Statistical Analysis".

We used SAS 9.4 (SAS Institute Inc. 2010, Cary, NC, USA) for data management and descriptive analyses. Machine learning analyses conducted using Python 3.9.7 (Python Software Foundation, 2021). This information has been provided in the methods section.

It should be explained how the qualitative and quantitative data are summarized under the sub-heading of Statistical Analyses in the Materials and Methods section of the study. Data analysis or Statistical analysis sub-section title should be added to the Materials and Methods.

We have included a section “Descriptive analyses” to explain how the data were summarized. The machine learning analyses section has details on how the data are summarized.

The exact P values should be added to the table(s) (e.g., p=0.25; p=0.03).

Table 2 provides descriptive statistics only; no hypothesis tests were performed and therefore no p-values are reported.

Table 3 now provides p-values for differences in pain drug use before and after diagnosis. The differences were tested with the McNemar test, a non-parametric method suitable for multiple observations on the same individual.

Table 4 displays all p-values (4 decimal points). However, any p-value smaller than 0.0001 is represented as <0.0001.

Are the data subjected to pre-processing? What approaches were used to test the validity of the models? Which metrics were used in the performance evaluation of the estimates of models/algorithms? How were the predictive models selected in this study? Which methods are used to model relationships between variables? The descriptions and other descriptive values/data should be defined on the tables and shapes.

The revised section on machine learning analyses explains all the steps that were used such as data pre-processing, model development, model performance, and model interpretation. We hope these descriptions will be helpul.

How were extreme/outlier values in the data determined and resolved?

All continuous predictors were standardized to zero mean and unit variance (StandardScaler). This z-score normalization places each feature on a common scale and helps to compress the influence of any residual extreme values. Because we then modeled with XGBoost—an ensemble of decision trees that is itself robust to outliers—no additional outlier exclusion was required.as XGBoost’s tree‐based splits are minimally affected by extreme values.

Thank you again for this opportunity to revise and resubmit our work. We have made additional edits to improve the readability of our manuscript.

We hope that our revisions meet the expectations of the reviewers and editorial team. Please let us know if you have any further comments, suggestions, or modifications.

Regards,

Dr. Rafia Rasu

---

## [Decision Letter · Decision Letter 1]

16 Nov 2025

Leading Predictors and their Associations with Combination Opioid Pain Therapy in Older Adults with Cancer: Application of Machine Learning Approaches

PONE-D-24-48620R1

Dear Dr. Rasu,

We’re pleased to inform you that your manuscript has been judged scientifically suitable for publication and will be formally accepted for publication once it meets all outstanding technical requirements.

Kind regards,

Sara Mucherino

Academic Editor

PLOS ONE

Additional Editor Comments (optional):

Reviewers' comments:

Reviewer's Responses to Questions

**Comments to the Author**

Reviewer #2: (No Response)

2. Is the manuscript technically sound, and do the data support the conclusions?

Reviewer #2: (No Response)

3. Has the statistical analysis been performed appropriately and rigorously?

Reviewer #2: (No Response)

4. Have the authors made all data underlying the findings in their manuscript fully available?

Reviewer #2: (No Response)

5. Is the manuscript presented in an intelligible fashion and written in standard English?

Reviewer #2: (No Response)

Reviewer #2: Accept

Accept

Accept

Accept

Accept

Accept

**Do you want your identity to be public for this peer review?** For information about this choice, including consent withdrawal, please see our Privacy Policy

Reviewer #2: No

---

## [Editor Report · Acceptance letter]

PONE-D-24-48620R1

PLOS One

Dear Dr. Rasu,

I'm pleased to inform you that your manuscript has been deemed suitable for publication in PLOS One. Congratulations! Your manuscript is now being handed over to our production team.

Kind regards,

on behalf of

Dr. Sara Mucherino

Academic Editor

PLOS One